computed tomography; deep learning; machine learning; medical imaging; cancer

**Author for correspondence:**
William C. McGough,
Email: wcm23@cam.ac.uk

# Artificial intelligence for early detection of renal cancer in computed tomography: A review

William C. McGough[1,2] , Lorena E. Sanchez[3,4], Cathal McCague[1,3,4], Grant D. Stewart[4,5], Carola-Bibiane Schönlieb[6], Evis Sala[3,4] and Mireia Crispin-Ortuzar[1,2]

[1]Cancer Research UK Cambridge Institute, University of Cambridge, Cambridge, UK; [2]Department of Oncology, University of Cambridge, Cambridge, UK; [3]Department of Radiology, University of Cambridge, Cambridge, UK; [4]Cancer Research UK Cambridge Centre, Cambridge, UK; [5]Department of Surgery, University of Cambridge, Cambridge, UK and [6]Department of Applied Mathematics and Theoretical Physics, University of Cambridge, Cambridge, UK

## Abstract

Renal cancer is responsible for over 100,000 yearly deaths and is principally discovered in computed tomography (CT) scans of the abdomen. CT screening would likely increase the rate of early renal cancer detection, and improve general survival rates, but it is expected to have a prohibitively high financial cost. Given recent advances in artificial intelligence (AI), it may be possible to reduce the cost of CT analysis and enable CT screening by automating the radiological tasks that constitute the early renal cancer detection pipeline. This review seeks to facilitate further interdisciplinary research in early renal cancer detection by summarising our current knowledge across AI, radiology, and oncology and suggesting useful directions for future novel work. Initially, this review discusses existing approaches in automated renal cancer diagnosis, and methods across broader AI research, to summarise the existing state of AI cancer analysis. Then, this review matches these methods to the unique constraints of early renal cancer detection and proposes promising directions for future research that may enable AI-based early renal cancer detection via CT screening. The primary targets of this review are clinicians with an interest in AI and data scientists with an interest in the early detection of cancer.

## Impact statement

Initially, this review discusses existing approaches in automated renal cancer diagnosis, and methods across broader AI research, to summarise the existing state of AI cancer analysis. Then, this review matches these methods to the unique constraints of early renal cancer detection and proposes promising directions for future research that may enable AI-based early renal cancer detection via CT screening.

## Introduction

In 2017, 393,000 renal cancer (RC) diagnoses and 139,000 RC deaths were recorded worldwide (Fitzmaurice et al., 2019). Renal cell carcinoma (RCC), the most common cancer involving the kidney, is mostly discovered incidentally during routine health checks or in the assessment of unrelated symptoms, and patients with incidentally discovered RCC tend to have better health outcomes than those diagnosed with symptomatic RCC (Rabjerg et al., 2014; Vasudev et al., 2020). This is because symptom presentation is generally associated with later-stage progression (Rabjerg et al., 2014; Vasudev et al., 2020). As shown in Table 1, RC screening satisfies many of the 10 Wilson–Junger criteria of an effective screening program (Wilson et al., 1968; Rossi et al., 2018); in principle, regular RC screening could improve general survival rates by increasing the rate of early RC discovery.

However, there are significant challenges associated with deploying the current standard method for RC discovery, contrast-enhanced computed tomography (CECT; Ljungberg et al., 2015; Guidelines for the Management of Renal Cancer, 2016), in RC screening: the high cost of computed tomography (CT) screening (Beinfeld et al., 2005; Ishikawa et al., 2007; Jensen et al., 2020), the risks of routine radiation exposure (Hunink and Gazelle, 2003), the lack of a definite target screening population (Rossi et al., 2018), and the low incidence of RC in the general population (O'Connor et al., 2011, 2018). These facts undermine LDCT's cost-effectiveness and suitability for ongoing screening – Wilson–Junger criteria 9 and 10, respectively. Nevertheless, recent literature has indicated that cancer screening with low-dose computed tomography (LDCT) may improve population health and studies are ongoing in this area (NLST, 2011; Black

**Table 1.** The current state of satisfaction of Wilson–Junger criteria for AI RC screening in LDCT

| Wilson–Junger criterion | Satisfied by AI-based RC screening in LDCT (Y/?/N) | Justification |
|---|---|---|
| 1. Tackles an important health problem | Y | Roughly 139,000 RC deaths in 2017 (Fitzmaurice et al., 2019) |
| 2. Can be followed by an acceptable treatment | Y | Treatment for RC is long-accepted and guideline-based (Ljungberg et al., 2015; Guidelines for the Management of Renal Cancer, 2016) |
| 3. Facilities for diagnosis and treatment are available | Y | Patients treated with earlier diagnosis have higher survival rates; malignancy-diagnosing needle biopsies are highly accurate (Rabjerg et al., 2014) |
| 4. A recognisable early stage exists | Y | 55–60% of all RC diagnoses are incidental (Rabjerg et al., 2014; Vasudev et al., 2020); RC initially exists in slow-growing latent state for many patients (Zhang et al., 2016) |
| 5. Suitable test or examination are available | N | Development of AI technology for LDCT screening is required |
| 6. Test is acceptable to population | ? | LDCT (among other screening methods) appears to be acceptable in screening the general population (Harvey-Kelly et al., 2020; Freer-Smith et al., 2021) |
| 7. Natural history of disease is well understood | N | The natural history of early RC is only partially understood (Volpe et al., 2004; Zhang et al., 2016; Rossi et al., 2018) |
| 8. Agreed policy on treatment | Y | Treatment policies for RC are well established (Ljungberg et al., 2015; Guidelines for the Management of Renal Cancer, 2016) |
| 9. Acceptable cost-effectiveness | ? | AI technology is not yet developed; high costs are likely in manual CT screening (i.e. without AI) (Beinfeld et al., 2005) |
| 10. Test is suitable as a continuing process | ? | NLST reduced participant mortality over 7-years of continuous screening; no equivalent study has been performed for RC (NLST, 2011) |

*Note.* Y: Yes, currently satisfied; ?: Unknown, more research is needed to clarify; N: No, currently unsatisfied.

et al., 2014; Stewart, 2021). Furthermore, developments in artificial intelligence (AI) have enabled the automation of some radiological tasks that may reduce the cost of CT analysis. Following these developments, this manuscript reviews AI technologies across automated RC diagnosis, other cancer domains, and broader computer vision to suggest novel research directions that may enable RC early detection in LDCT and non-contrast CT (NCCT), by automating and reducing the cost of analyses inherent to CT screening.

In this review, we define 'early detection' as the processes requisite in screening that detect early signs of disease in asymptomatic individuals. Image-based early detection and diagnosis may share many sub-processes, such as pre-processing, segmentation, radiomic feature extraction, post-processing, and classification. Within these sub-processes, segmentation and classification are the subjects of most machine learning research. Segmentation algorithms receive images as input and assign to them element-wise labels according to predefined semantic values, providing structure to images by highlighting the most salient regions of interest (ROI), making automated analyses simpler. An example of two-dimensional segmentation is shown in Figure 1. Classification refers to any process that assigns a discrete category to a data source; classification algorithms receive quantitative data (e.g., radiomic features, morphological measurements from a histology slide, or raw pixel data from an image) and assign a label to the data source; this label can be binary (malignant/benign) or multi-class (differentiating between RCC subtypes).

Early detection methods must be cheap to be viable in screening. They must also be accurate, to detect a high rate of the target disease whilst minimising the rate of overdiagnosis, which can dramatically increase screening costs. AI analyses are automated by default, making them cheap enough to be operationally viable in screening. Therefore, the development of an AI-based RC early detection system should focus on optimising the AI system's accuracy to maximise the system's utility in screening.

This manuscript reviews existing AI diagnostic methods that may be suitable for early detection, and suggests possible improvements to these existing methods, due to the lack of existing AI research in RC early detection. The literature reviewed in this manuscript was extracted from three different sources, namely (i) Kidney and Tumour Segmentation Challenge (KiTS) winning submissions; (ii) ImageNet (March 2022), including four contemporary, high-scoring algorithms and four other highly cited algorithms often used in medical AI, and (iii) renal segmentation and classification articles (Google Scholar, January 2015–March 2022). A list of all papers initially selected for reading, and then finally included, in this review can be found in the Supplementary Material. The review is complemented by highly cited articles from other early detection domains, that may represent novel approaches for conducting AI LDCT screening for RC, and the broader AI literature, including hyperparameter optimisation, multi-task learning (MTL), and synthetic image generation.

### AI primer

AI refers to any computational, data-driven decision-making system that enables the automation of complex tasks – mimicking human intelligence – without explicit instruction. Machine-learning models are a subset of AI systems that automatically learn

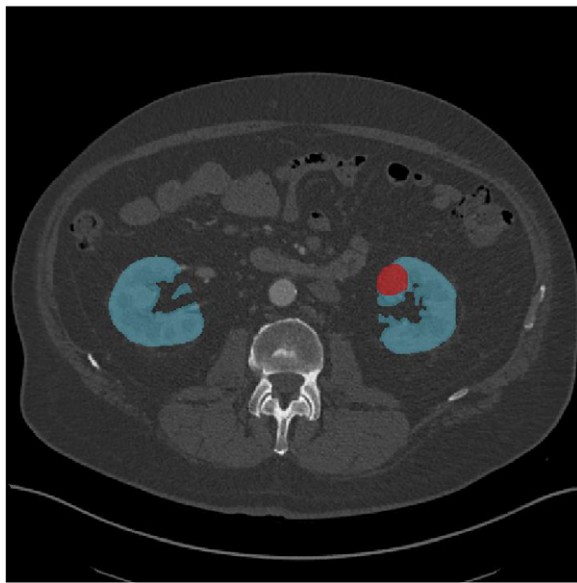

**Figure 1.** A segmented CECT axial slice, depicting the segmented kidneys (blue) and tumour(red). CT data taken from KiTS19, case 49.

ratio of these response classifications, such as sensitivity and specificity,

$$\text{Sensitivity} = 100\left(\frac{n_{\text{TP}}}{n_{\text{TP}} + n_{\text{FN}}}\right), \tag{1}$$

$$\text{Specificity} = 100\left(\frac{n_{\text{TN}}}{n_{\text{TN}} + n_{\text{FP}}}\right), \tag{2}$$

where $n_x$ refers to the number of $x$ observed in validation. Optimum performance usually requires a trade-off between maximum specificity and maximum sensitivity; the area under the receiver operating characteristic curve (AUC) and the Dice similarity coefficient (DSC) are commonly used accuracy metrics that quantify the model's trade-off between specificity and sensitivity. AUC is generated by plotting the model's receiver operating characteristic (specificity vs. sensitivity) and calculating the area under its curve; an example ROC is shown in Figure 2 for the reader's understanding. Segmentation performance is generally evaluated by the DSC metric, defined by

$$\text{DSC} = \frac{2n_{\text{TP}}}{2n_{\text{TP}} + n_{\text{FP}} + n_{\text{FN}}}. \tag{3}$$

to structure and/or make predictions, or 'inferences', from data. Supervised learning models learn using *labelled* datasets – a set of paired inputs and labelled outputs. In segmentation, labelled datasets contain CT scans and volumes of corresponding voxel-wise labels for each scan. Supervised learning models review labelled data during 'training', iteratively assessing each sample and altering its own mathematical parameters to progressively improve inference accuracy. Following training, a supervised learning model's accuracy is evaluated over an unseen 'validation' labelled dataset, where the differences between the model's inferences and the dataset's labels are evaluated to determine the model's overall accuracy. This manuscript exclusively reviews supervised machine-learning methods but, for brevity, 'AI' will be used as a general term for all models.

In classification and segmentation, the model's responses can be categorised as true positive (TP), true negative (TN), false positive (FP), or false negative (FN). Accuracy metrics are derived from the

Contemporary AI algorithms in image analysis tend to be comprised of convolutional neural networks (CNN) and/or transformers. This manuscript will not discuss the technical differences between these models, beyond the functional differences that exist with respect to their typical performance and cost characteristics. Both are deep learning algorithms (DL), meaning they are both types of neural network. The cost of CNNs scales linearly with the number of input image elements, whereas transformer cost scales quadratically, making transformers-only models much costlier during analyses of 3D images, such as in CT. Transformers can achieve 'global' attention and detect patterns across whole input images simultaneously, whereas CNNs can only achieve 'local' pattern recognition, as they must divide input images into smaller sections and analyse them individually. This leads to superior image analysis in transformer models where patterns have global interdependencies. The performance of CNN- and transformer-based models will be reviewed in this manuscript, as well as hybrid models that attempt to combine the benefit of both approaches.

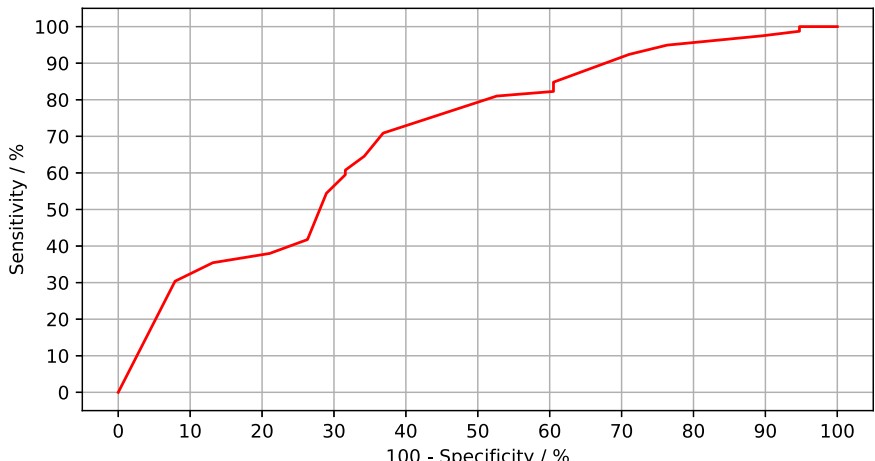

**Figure 2.** An example ROC curve for an arbitrary classifier, displaying the trade-off between sensitivity and specificity in an arbitrary classification task. The further the curve is from the x-axis, and the closer it is to the y-axis, the higher the classifier's holistic accuracy and AUC. In the shown ROC curve, AUC is 0.699.

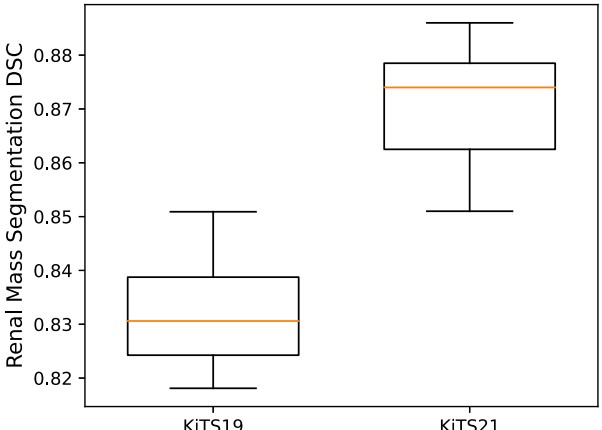

**Figure 3.** The performance distribution of the top-7 algorithms in KiTS19 and KiTS21, with respect to mass segmentation DSC. Due to the labelling differences between KiTS19 and KiTS21, all masses in KiTS19 are labelled as 'Tumour', whereas masses in KiTS21 are labelled as either 'Tumour' or 'Cyst'.

## AI in renal cell carcinoma diagnosis

### Segmentation

Renal segmentation has received increased research attention following the advent of KiTS, first established in 2019 (Heller et al., 2019) and renewed in 2021. KiT19 and KiTS21 publicly released 210 CECT volumes and 300 CECT volumes, respectively, where all CT scans contained tumours and some contained cysts, and invited participants to submit their renal segmentation algorithms to compete in a fair assessment of accuracy. KiTS19's winner, based on nnU-Net (Isensee and Maier-Hein, 2019; Isensee et al., 2021), was derived from the state-of-the-art segmentation CNN U-Net (Ronneberger et al., 2015) and focused on the optimisation of its hyperparameters (properties relating to training and model size) without altering the essential structure of U-Net. This approach represented a breakaway from the hitherto standard across segmentation research, of proposing modular architectural changes to U-Net for marginal accuracy gains. Outside of KiTS, nnU-Net scored highly in a wide variety of segmentation domains, winning other medical segmentation competitions across multiple organ sites (Isensee et al., 2021), proving the primacy of hyperparameter optimisation in maximising segmentation performance.

All KiTS21's top-7 performing submissions made direct use of nnU-Net as a baseline algorithm. The top-3 submissions used nnU-Net's 'course-to-fine' cascade approach. In this approach, a 'course' U-Net segments the input CT images at a low resolution to dictate an initial ROI; then, this segmentation inference is refined at higher resolutions by more 'fine' U-Nets. This process is repeated until the ROIs are labelled at full resolution. Figure 3 shows the performance distributions of KiTS19 and KiTS21's top-7 submissions in renal segmentation – the adoption of nnU-Net significantly increased the mean mass-segmentation DSC among top performers ($p = 3.58 \times 10^{-5}$) from 0.832 in KiTS19 to 0.870 in KiTS21 (Challenge Leaderboard, 2019; KiTS21, 2021). To the authors' knowledge, no other kidney segmentation algorithm has significantly improved upon KiTS21's competition-winning nnU-Net-based approach (Zhao et al., 2022).

Manual NCCT screening exhibits potential as a medium for RC early detection (O'Connor et al., 2011, 2018), yet there has been little supporting research in NCCT segmentation that may assist the automation of NCCT screening. LDCT and NCCT images are

significantly noisier and less differentiated than in CECT, respectively, making target organs harder to distinguish for AI algorithms. Transference of segmentation algorithms between the CECT and NCCT or LDCT may be non-trivial due to the differences in image quality, thus new work must quantify the performance of segmentation within NCCT images, to verify the suitability of segmentation-based RC early detection in NCCT.

### Classification

Renal classification algorithms generally fall into one of the following characterisations: DL-based (Han et al., 2019; Tabibu et al., 2019; Fenstermaker et al., 2020; Oberai et al., 2020; Pedersen et al., 2020; Tanaka et al., 2020; Zabihollahy et al., 2020; Uhm et al., 2021), feature analysis-based (Hodgdon et al., 2015; Schieda et al., 2015; Feng et al., 2018; Kocak et al., 2018; Lee et al., 2018; Schieda et al., 2018; Varghese et al., 2018; Erdim et al., 2020; Ma et al., 2020; Sun et al., 2020; Wang et al., 2021), or a hybrid approach (Lee et al., 2018; Tabibu et al., 2019). The higher inference time and cost of DL-based algorithms compared to feature-based algorithms is undesirable, but DL-based approaches tend to be more accurate.

DL-based classification approaches generally use 'fine-tuned' versions of pretrained CNN classifiers (such as ResNet, He et al., 2016; VGG, Simonyan and Zisserman, 2015; or Inception, Szegedy et al., 2016). Fine-tuning in this context means to retrain an already existing pretrained model to operate effectively in a new domain. This approach minimises the need for domain-specific labelled images (and, therefore, minimises labelling), and provides good classification performance. Feature-based algorithms operate on predetermined ROIs – image sections segmented by a radiologist or AI algorithm – and use radiomic and/or DL-derived features, that describe relationships in the local distribution of CT intensities, to classify disease.

Deep learning-based classifiers can achieve high accuracy in CT images with very little manual intervention. Tanaka et al. (2020) sought to quantify small (≤4 cm) renal mass detection accuracy in CT using axial CT slices and a fine-tuned InceptionV3 CNN; they differentiated malignant and benign masses with a maximum AUC of 0.846 in CECT and 0.562 in NCCT. Pedersen et al. (2020) trained a similar 2D slice-classifying CNN, but used it to classify each slice within each known mass' 3D volumes to enable a slice-based voting system to differentiate patient-level RC from oncocytoma, returning a perfect validation accuracy of 100%. Han et al. (2019) sought to differentiate between clear cell RCC (ccRCC) and non-ccRCC from known RCC masses, using radiologist-selected axial CT slices from NCCT and two CECT phases, and achieved sub-type classification AUCs between 0.88 and 0.94 in an internal testing dataset.

Classification has also been performed with the following feature-based supervised learning models: support vector machines (SVM; Hodgdon et al., 2015; Schieda et al., 2015; Kocak et al., 2018; Erdim et al., 2020; Sun et al., 2020), multi-layer perceptrons (MLP; Kocak et al., 2018; Erdim et al., 2020), logistic regressions (LR; Hodgdon et al., 2015; Schieda et al., 2015; Schieda et al., 2018; Varghese et al., 2018; Ma et al., 2020; Wang et al., 2021), and decision tree methods (DT; Lee et al., 2018; Erdim et al., 2020). Some feature-based models have shown superior diagnostic performance to expert radiologists: Hodgdon et al.'s (2015) SVM-based approach classified RC in NCCT images with an AUC of around 0.85; this was much greater than the radiologists' AUCs of 0.65 and 0.74. Sun et al.'s (2020) 'radiologic-radiomic' SVM model, where 'radiologic' refers to human-derived radiographic features and 'radiomic' refers to machine-derived radiographic features,

differentiated RCC subtypes from benign masses. Sun et al. (2020) reported their accuracies in DSC, achieving an average of 88.3% DSC, improving upon the 78.2% average expert radiologist's DSC (individual radiologists varied between 73.2 and 84.1%).

Across RC classification literature, the interaction between feature analysis and DL models is limited. Tabibu et al.'s (2019) classification pipeline sends patches of histopathological images to two CNNs – one CNN classifies each patch as benign/malignant, and the other generates features that are used to differentiate between RCC subtypes in a three-class SVM. In internal validation, performing classification on histopathological images, this method achieved up to 0.99 patch-wise malignancy-identification AUC, and 0.93 subtype-identification AUC. Lee et al.'s (2018) approach concatenated radiomic features with a CNN output, both evaluated over a pre-segmented ROI in a CT image and fed this concatenation to a DT classifier that differentiated angiomyolipoma without visible fat from RC with up to 0.816 AUC.

Object detection has rarely been applied to renal mass detection in CT (Yan et al., 2018; Xiong et al., 2019; Zhang et al., 2019). Zhang et al.'s (2019) renal lesion detector show a mass-level detection AUC of 0.871 in CECT; they did not compare this performance to expert radiologist performance over the same validation dataset. As in segmentation, the reduced image quality of NCCT may present issues for AI lesion detection algorithms; thus, to ensure suitability in early detection, work must be done to quantify object detection performance in NCCT.

### MTL and synthetic image generation

AI has been used to support RC diagnosis in other interesting manners, including MTL and synthetic image generation (SIG). SIG aims to create new images that mimic the appearance of authentic medical images. In RC, SIG has been used to improve segmentation performance (roughly 0.5% DSC improvement, Jin et al., 2021) by synthetically expanding the size of labelled training datasets, and shows promise in improving classification performance by synthetically transferring images to more diagnostically-useful domains, such as from NCCT to CECT (Liu et al., 2020; Sassa et al., 2022). However, to the authors' knowledge, no research has quantified the improvement in RC classification performance directly attributable to synthetic domain transfer between NCCT and CECT. MTL has been used in RC evaluation to combine learning from multiple tasks, such that they simultaneously contribute towards model training – Ruan et al. (2020) noted a 3% segmentation DSC improvement following MTL, and Pan et al. (2019) noted how classification and segmentation performance scores were both individually improved when trained together in MTL.

### Alternate methods of using medical AI

### Alternate detection paradigms

Rather than removing the need for pathologist personnel in screening, Gehrung et al.'s (2021) AI approach generated a proxy 'confidence' rating to triage patients suspected of having Barrett's oesophagus, a precancerous state for oesophageal cancer. Their AI detected 'indeterminate' cases and sent these to an expert pathologist, whilst accurately assigning classifications to 'clear' cases. Gehrung et al.'s (2021) triage approach was rigorously assessed across multiple validation datasets and was estimated to reduce pathologist workloads by 57% without a reduction in accuracy, improving the cost-effectiveness of screening. As in Barret's oesophagus, triaging AI may be practicable in LDCT RC screening and improve the process' cost-effectiveness (Wilson–Junger criterion 8, Table 1).

Khosravan et al. (2019) found that humans tend to have higher specificity and AI algorithms tend to have higher sensitivity in NCCT lung cancer detection; in response, they constructed a 'complimentary' computer-aided diagnosis system to bridge the performance gap between radiologists and AI. Khosravan et al.'s (2019) system let a radiologist evaluate an input NCCT image as the AI system segmented and classify each gaze-deduced region of interest, generated by the radiologist's eye movement, automatically. This study failed to specify the improvement in cancer detection, or workload reduction, directly attributable to their software, instead plainly evaluated the performance of segmentation (91% DSC) and classification (97% accuracy – AUC not reported).

### Object detection in AI cancer detection

Ardila et al. (2019) used an object-detection algorithm to identify lung nodules in NCCT with high accuracy, allowing patient-level early cancer detection AUC of 0.944. Welikala et al. (2020) used an object detection algorithm to identify oral lesions in plain photographic images of the oral cavity, allowing patient-level cancer classification, and achieving a patient-level classification DSC between 78 and 87% (AUC not reported). Nguyen et al. (2022) proposed a circular 'bounding-box' object detection algorithm for general biological purposes, as certain biological structures tend to be more circular/spherical than rectangular/cuboidal such as cells, masses, and some organs. They proved that their 'CircleNet' object-detection algorithm showed overall superior performance to other state-of-the-art algorithms in detecting nuclei and glomeruli.

### Synthetic image generation

Santini et al.'s (2018) DL workflow synthetically enhance NCCT images, promoting them to pseudo-CECT, to enable accurate estimation of patient cardiac volumes. Santini et al. (2018) proved the efficacy of this method by highlighting the segmentation improvement associated with synthetic CECT generation; their framework, performing segmentation over synthetic CECTs, was more accurate than a human over an equivalent set of NCCTs (DSC of 0.89 and 0.85, respectively). Hu et al. (2022) built a generative adversarial network (GAN) to generate realistic synthetic CECT images that improve the conspicuity of abdominal aortic aneurysms in NCCT images. Their GAN made use of U-Net to generate synthetic CECT images, and was trained in MTL – using vascular structure segmentation as an auxiliary task to boost the performance of CECT generation. Hu et al. (2022) found that their GAN outperformed stand-alone U-Net, and other SIG algorithms such as pix2pix (Isola et al., 2017) and MW-CNN (Liu et al., 2018), in terms of average validation error and signal-to-noise ratio. Qualitatively, Hu et al. (2022) showed clearly that the noise produced in U-Net-based NCCT to CECT translation is minimised by its incorporation into a GAN. Hu et al. (2022) did not directly quantify the improvement in aneurysm detection directly attributable to their synthetic CT enhancement, but they did determine case-level aneurysm detection DSC to be 85%.

## Emergent ideas across AI and computer vision

### Segmentation

Yang et al. (2022) found that exhaustive hyperparameter optimisation of large AI models, such as CNNs and transformers, is possible – they showed neural networks over a very large range of sizes can share common optimal hyperparameters if they are initialised 'correctly'. This correct initialisation allows grid-search-based objective hyperparameter optimisation, which nnU-Net established as primarily important in segmentation. Also, the intrinsic locality of convolutional operations in CNNs may limit U-Net's performance in segmentation tasks with global pattern dependencies. Introducing transformers, capable of global attention and understanding the relationships between all input data, to the U-Net architecture may allow the model to 'see' much larger volumes during segmentation, which may improve segmentation accuracy. TransU-Net and UNETR both implemented transformers into U-Net's CNN architecture and significantly improved upon U-Net's segmentation performance in multi-organ segmentation tasks (Chen et al., 2021; Hatamizadeh et al., 2022).

### Classification

Following the introduction of transformers (Vaswani et al., 2017; Dosovitskiy et al., 2020), a new generation of state-of-the-art classifiers (including ConvNeXt, Liu et al., 2022), Swin (Liu et al., 2021) and CoaT (Xu et al., 2021), have superseded the commonly used CNNs Resnet, VGG and Inception in terms of ImageNet classification accuracy. This new generation shows improved performance over the same tasks due to their new training regimes, new hyperparameters and new architectures. ConvNeXT (which, like the previous generation of classifiers, is a pure CNN) tweaked its properties to take advantage of insights made by transformers models (Liu et al., 2022) and shows improved performance over the previous generation without incurring greater cost during inference.

### Multi-task learning

Standley et al. (2020) assessed various methods of combining AI training regimes. They found that some 'complex' tasks, such as segmentation, require greater number of training samples for optimal performance than other 'simpler' tasks, and that these more complex tasks' performances would suffer if paired with a simple task in MTL. Standley et al. (2020) also found that some tasks seemed to consistently act as 'auxiliaries' – boosting the learning performance of the network for other tasks without ever performing significantly well themselves in MTL. Despite these findings, they found that the relationships between task pairings – that is, the tendency of tasks to help or hinder each other's training during MTL – was not independent of the training setup, meaning MTL relationships between tasks cannot be completely generalised across models with distinct network architectures, hyperparameters, and training data.

## Discussion

Renal segmentation has the potential in assisting RC diagnosis – for example, accurately delineating tumour regions enables feature-based classification, which shows comparable, or superior, diagnostic performance to expert radiologists. Maximising renal segmentation accuracy in LDCT may enable accurate feature-based classification methods to be applied in LDCT early detection automatically, removing much of the manual labour of RC screening. High accuracy is essential in early detection methods; thus, given the accuracy of the feature-based classification methods in NCCT imaging (as in Hodgdon et al., 2015), a high-accuracy renal segmentation method for LDCT is likely to enable RC early detection screening.

Whilst nnU-Net established the primacy of hyperparameter optimisation in segmentation performance, it does not provide a framework for hyperparameter optimisation itself, instead relying on experimentally derived heuristics for hyperparameter selection. Using Yang et al.'s (2022) 'maximal parameter update' hyperparameter optimisation allows a definitive optimisation of any CNN or transformer, which should improve upon nnU-Net's heuristics-led approach. Also, despite nnU-Net's state-of-the-art inter-domain performance, the intrinsic locality of convolutional operations in U-Net's purely convolutional architecture may limit its segmentation performance. Introducing transformers to U-Net's architecture, as in TransU-Net, enables global attention mechanisms that may improve RC segmentation accuracy over a whole NCCT volume. Applying transformer-informed segmentation methods like TransU-Net, and objectively optimising its hyperparameters using 'maximal parameter updates' may improve RC segmentation performance over existing nnU-Net-led approaches.

Given the potential for RC early detection in LDCT, there is a need for more research quantifying RC segmentation performance in LDCT. Investigations into general NCCT segmentation have shown that using synthetic contrast enhancement as an auxiliary training task in MTL can improve segmentation accuracy. Therefore, an investigation in renal LDCT segmentation may be improved by introducing synthetic enhancement to CECT as an auxiliary learning task in MTL. Such an investigation would likely be complicated by Standley et al. (2020) findings – that MTL task relationships can be unique to each configuration of network architecture, hyperparameters, and dataset domain.

Like segmentation, the lack of research quantifying RC object detection performance in LDCT represents a gap in the literature. Object detection and classification performance could be improved by the introduction of the new generation transformer-inspired classifiers that consistently show higher classification accuracies than their predecessors. Also, assessing the MTL relationship between classification, segmentation, and object detection in RC early detection may lead to improved mass detection, and therefore early detection, performance.

Pedersen et al.'s (2020) and Gehrung et al.'s (2021) approach of generating an image-based intra-patient biomarker voting system may be applicable to RC early detection. Both Pedersen et al. (2020) and Gehrung et al. (2021) evaluated biomarker presence in fractionated tiles of input images and used the ratio of biomarker-positive to biomarker-negative tiles to classify the inputs, leading to high-accuracy results in validation. Applying an analogous approach, using the new generation of classifiers, to the early detection of RC masses in LDCT could enable highly robust automated triaging, or diagnosis, for RC early detection screening programmes.

## Conclusion

This manuscript highlights and summarises existing AI method in RC diagnosis and suggests how these can be repurposed to enable

RC early detection. After summarising existing segmentation, classification, and other AI methods in RC diagnosis, a review of analogous cancer detection and diagnosis methods across broader cancer literature and computer vision was conducted. Contrasting the RC-specific workflows to their equivalents across computer vision and other cancer domains allowed the generation of novel RC-specific research proposals that may enable AI-based RC early detection.

**Open peer review.** To view the open peer review materials for this article, please visit http://doi.org/10.1017/pcm.2022.9.

**Supplementary material.** To view supplementary material for this article, please visit https://doi.org/10.1017/pcm.2022.9.

**Financial support.** This work was supported by the International Alliance for Cancer Early Detection, a partnership between Cancer Research UK (C14478/A27855), Canary Center at Stanford University, the University of Cambridge, OHSU Knight Cancer Institute, University College London and the University of Manchester. This work was also supported by the CRUK National Cancer Imaging Translational Accelerator (NCITA) (C42780/A27066), and The Mark Foundation for Cancer Research and Cancer Research UK (CRUK) Cambridge Centre (C9685/A25177). Additional support has been provided by the Wellcome Trust Innovator Award, UK (215733/Z/19/Z) and the National Institute of Health Research (NIHR) Cambridge Biomedical Research Centre (BRC-1215-20014). The views expressed are those of the authors and not necessarily those of the NHS, the NIHR or the Department of Health and Social Care.

**Competing interest.** The authors of this manuscript declare relationships with the following companies: E.S. is a co-founder and shareholder of Lucida Medical Ltd. L.E.S. has received consulting fees from Lucida Medical Ltd. The remaining authors declare that they have no conflicts of interest to declare.

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
