## [Reviewer Report]

*Comments to Author*: This review deals with advances in AI for radiological early detection of renal cell carcinoma (RCC). The focus is on technical aspects and the developments therein, whereas the route to implementation and the role it can have in screening is superficially dealt with. The title is not fully in line with the scope.

This review is interesting mainly for readers who are interested in technical aspects of the use of AI in radiology

Remarks:

- although this is not a systematic review, still some indication on the approach to find and select articles are needed

- the authors describe that many of the 10 Wilson-Junger criteria are met, yet give examples that are not met. I am doubtful that CT for early RCC is really close to implementation; a table might be helpful

- it is stated that for screening te procedure needs to quick, with rapid reporting: for screening this is less important for clinical questions; in fact some actual screening methods like for colorectal cancer and cervical cancer are not that quick

- the term classification is used in two different situations: radiological and pathological. This leads to confusion: in the one case it is tissue/tumor separation in the other it is the categorization into tumortype

- some examples of application of AI in clinical practice are given, including in pathology. Although there are several articles, there is very little implementation in pathology practice, in fact probably only in the field of lymph node evaluation for metastasis, which is not mentioned in the review. Furthermore, there is literature on the use of AI in radiology in lung and breast cancer screening that gets very little attention

---

## [Editor Report]

*Comments to Author*: This manuscript gives an extensive and technical overview of repurposing existing AI approaches for RCC early detection, ending with recommendations to improve both segmentation and classification approaches to enable early RCC detection.

This is a well-written review of the literature and I believe will be well received by the community.

A few suggestions:

1) I suggest combining subsections 3.3 and 3.4 to be consistent with section 2. 

2) I suggest including a table summarising each reference referred to in sections 3 and 4 would support the reader in navigating between referrals to the references in the Discussion with the main body of the text

3) I wonder if the title should be something like: 'ADVANCING EARLY DETECTION OF RENAL CANCER WITH ARTIFICIAL INTELLIGENCE IN COMPUTED TOMOGRAPHY'. As not all approaches in the paper are 'new'?

4) The abstract of the article and impact statement must be given in the article before the introduction.

In general, the paper was clear and adds to the knowledge in this field - I hope to see the recommendations made in this article taken forward.